# Feature-based image registration in structured light endoscopy

**Andreas M. Kist**[1,2,*]                    ANDREAS.KIST@FAU.DE
[1]*Department of Artificial Intelligence in Biomedical Engineering, Friedrich-Alexander-University Erlangen-Nürnberg, Germany*

[2]*Division of Phoniatrics and Pediatric Audiology, Department of Otorhinolaryngology, Head- and Neck surgery, University Hospital Erlangen, Friedrich-Alexander-University Erlangen-Nürnberg, Germany*

**Julian Zilker**[2,*]                        JULIAN.ZILKER@GMX.DE
**Michael Döllinger**[2]             MICHAEL.DOELLINGER@UK-ERLANGEN.DE
**Marion Semmler**[2]                  MARION.SEMMLER@UK-ERLANGEN.DE

**Editors:** Under Review for MIDL 2021

## Abstract

Images offer a two-dimensional (2D) representation of a three-dimensional (3D) environment. However, in many biomedical tasks, a 3D view is crucial for diagnosis. Projecting structured light, such as a regular laser grid, onto the surface of interest allows to reconstruct its 3D structure. For reconstruction, it is crucial to correctly identify and assign each laser ray to its respective position in the laser grid. Current methods for this task use semi-automatic, yet highly manual annotations. Hence, a fully automatic, reliable method is desired. Here, we show that this assignment can be approached as an image registration. After separating the laser rays from the background, we found that registration of the extracted laser rays directly to the fixed laser grid image fails, when we use state-of-the-art intensity-based image registration techniques, such as the Advanced Normalization Tools (ANTs). Using our feature-based custom loss and a deep neural network, we are able to use a U-Net-like architecture to compute deformation fields to successfully register the laser rays onto the fixed image accompanied with a custom post-processing assignment step. Using synthetic data, we show that the network is in general able to learn affine and non-linear transformations. Our method is also robust to missing or occluded rays. Using an *ex vivo* dataset, we achieved a registration accuracy of 91%. In summary, we provide a new platform to perform feature-based registration and showcase this on a biomedical dataset. In the future, we will evaluate different architectural designs and more complex datasets.

**Keywords:** Image registration, deformation field, structured light, endoscopy

## 1. Introduction

A three-dimensional (3D) view is highly important for an accurate biomedical diagnosis and treatment in many areas, such as MRI for strokes and CT for bone fractures. However, multiple professions still rely on imaging procedures that use conventional cameras, such as laryngeal endoscopy. Here, only a 2D view onto the larynx is available (Andrade-Miranda et al., 2020; Deliyski and Hillman, 2010). However, laryngeal endoscopy is the gold standard

---

* These authors contributed equally.

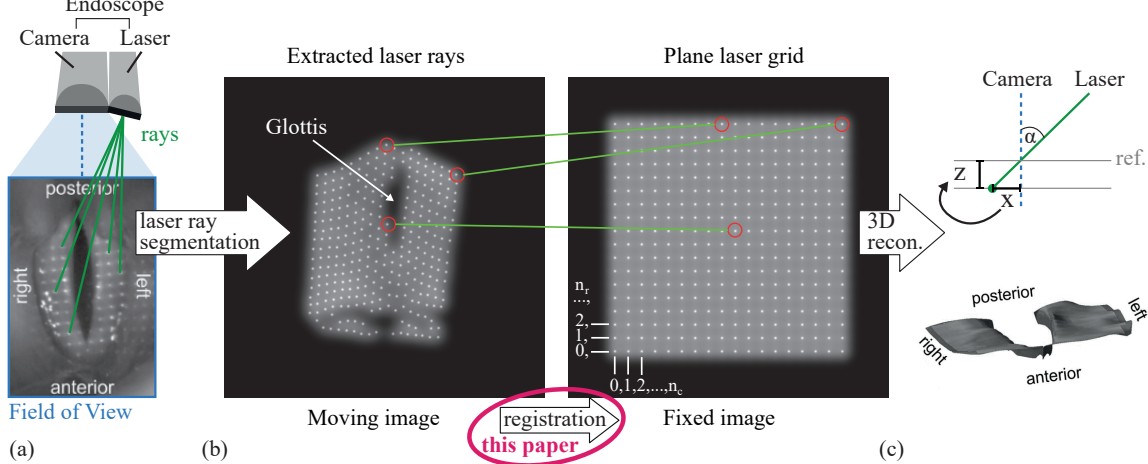

Figure 1: Structured light endoscopy. (a) An endoscope consisting of a camera and a laser projection unit. Laser rays produce a point grid that is directed onto the surface of interest and visible in the camera image. (b) The extracted laser rays have to be assigned to a unique grid position in the plane laser grid for valid 3D reconstruction. $n_c$ and $n_r$ describe the number of columns and rows, respectively. (c) 3D reconstruction by inferring the depth information $z$ by determining the distance in $x$ (and $y$, not shown for clarity) compared to the reference given a fixed camera to laser angle $\alpha$.

to assess the health state of a subject's voice (Mehta and Hillman, 2012). A healthy voice features a symmetric, homogeneous movement of the vocal folds in three dimensions (Titze and Martin, 1998), however, with the aforementioned technique, the examiner has only limited access to this information. Therefore, there is a drive to develop 3D endoscopic techniques (Luegmair et al., 2010; Ghasemzadeh et al., 2020; Semmler et al., 2016; Schmalz et al., 2012).

Recently, we and others showed that laryngeal endoscopy using structured light (Geng, 2011) is feasible for 3D reconstruction of the vocal folds (Ghasemzadeh et al., 2020; Semmler et al., 2016; Luegmair et al., 2010, 2015). By using laser rays in a fixed angle $\alpha$ in relation to the endoscope (Figure 1(a,c)), we can use triangulation for each laser ray to reconstruct the 3D surface. Briefly, in a calibrated laser grid, the depth is computed by the relative distance of each laser ray to its reference position (Figure 1(c)). The key issue here is the laser ray extraction and correct individual assignment, which remains very challenging and requires a significant amount of manual effort (Semmler et al., 2017). Only recently, there are first reports using deep learning in structured light endoscopy (Li et al., 2019; Ma et al., 2019).

In general, we hypothesized that this task can be approached as an image registration procedure (Hill et al., 2001) between a fixed (the ideal grid) and a moving image, i.e. the laser rays extracted from an endoscopic image (Figure 1(b)). There exist multiple registra-

tion platforms, such as the Computational Morphometric Toolkit (CMTK, Rohlfing 2011) and the Advanced Normalization Tools (ANTs, Avants et al. 2009) to register biomedical image data, such as individual MRI brain scans to a brain atlas. In general, deformable medical image registration is a complex, fast forward moving field with many strategies and applications to achieve a good registration (Sotiras et al., 2013), for example the use of phantoms to align different image modalities (Rodríguez-Ruano et al., 2008). Recently, a large body of deep neural networks have been used for (biomedical) image registration (Fan et al., 2019; Jaderberg et al., 2015; Yang et al., 2017; Krebs et al., 2019), which have been recently summarized (Haskins et al., 2020). Recent advances utilize the prediction of deformation fields using U-Net-like architectures (Balakrishnan et al., 2018), which can be combined with general adversarial networks (Mahapatra et al., 2018). Most of the architectures mentioned, however, were evaluated on tomographic images that contain a high-level structure that is non-repetitive. Further, these methods align images typically by minimizing an intensity-based metric. Although recent point-registration deep neural networks exist, they were only evaluated on point clouds that contain special features (Aoki et al., 2019) or find only rigid transformations (Wang and Solomon, 2019). In contrast, the laser grids are highly regular and repetitive, and are potentially hard to align only based on intensity and contain likely non-rigid transformations.

In this work, we are investigating if laser grid maps are able to be registered onto an ideal grid using well-established tools (ANTs) and deep neural networks. We test if intensity-based registration is able to accurately map individual laser rays and if a feature-based approach is more suitable. We develop therefore a custom loss function and test the registration ability on a synthetic dataset. We show the general applicability and performance on *ex vivo* data of oscillating vocal folds.

## 2. Methods

We provide the code for this study on GitHub https://github.com/julzil/endolas.

### 2.1. Synthetic dataset

We generated an evenly spaced laser grid containing 25 keypoints organized in five columns and five rows (5×5). The image size was 224×224 px for initial experiments to test what the network is capable to learn and for optimization strategies (as shown in Figure 3). We generated three sets containing each 4800 synthetic images plus one fixed image (no perturbations). For each set, we randomly applied a different set of perturbations (as shown in Figure B.1): The first set used affine transformations only (i.e. translation, rotation, shear, and scaling), the second one used affine and non-linear transformations (i.e. applying a sine), and the third one also included random keypoint dropout to mimick hidden, occluded or not extracted keypoints with a probability of 0.2. The sets were randomly split into 70% training, 15% validation and 15% test set. For *ex vivo* experiments, we adjusted the synthetic dataset accordingly: We changed the resolution to 768×768 px and used 324 keypoints arranged in an 18×18 grid.

### 2.2. *Ex vivo* dataset

Calf larynges were obtained from the local slaughter house and prepared as previously described (reviewed in Döllinger et al. 2011). All footage was recorded using a Photron Highspeed Camera at 4,000 fps at a resolution of 768×768 px. A 532 nm Nd:YAG laser was used for laser grid generation. The collimated laser light was focused via an 18×18 micro-lens array to yield a focal plane at around 80 mm below the light outlet. The laser grid was calibrated using a custom calibration script in MATLAB (Semmler et al., 2016). We analyzed twelve videos in total, each recorded on a unique calf larynx. Out of the twelve videos, eight videos belonged to the training, two to the validation and two to the test set. Each video contained twenty fully annotated frames. Therefore, the training set contained 160, the validation 40 and the test set 40 frames. In each frame, all visible keypoints were annotated manually and assigned to a unique grid position (Figure 1(b)) for evaluation. These annotations were used as ground truth.

### 2.3. Image generation

Keypoints and their $(x, y)$ location were either known by creation (synthetic dataset, see 2.1) or manually annotated (*ex vivo* dataset). The latter is typically retrieved automatically in a working environment using semantic segmentation of the laser rays, e.g. using a U-Net, with subsequent 2D peak finding. For each keypoint, we drew a small circle with a certain radius using the keypoint coordinates as centroid onto an initially black image with the same dimensions as the original image. We further added a Gaussian blur version of the image to incorporate a low-level structure to the image. This approach resulted in images as shown in Figure 1, Moving image.

### 2.4. Neural network architecture

We trained a modified U-Net architecture (Ronneberger et al., 2015) using $k = 32$ base filters implemented in TensorFlow/Keras (implementation from Gómez et al. 2020) in v.2.2.0 and 2.3.0, respectively. An overview is given in Figure 2. Each block contained a Conv2D layer with a 3×3 kernel, followed by a BatchNorm-Layer and a ReLU activation (similar to Çiçek et al. 2016). The final layer contained a 1×1 Conv2D layer with linear activation with two filters, resulting in two maps $u_x(\mathbf{x})$ and $u_y(\mathbf{x})$, for $x$ and $y$ translation, respectively. As input we used either only the moving image, the fixed and the moving image, or the moving image, the gradient and the difference image, as suggested by (Fan et al., 2019), see also Table 1. The input image size was 224×224 px and 768×768 px for synthetic and *ex vivo* data set, respectively. The fixed image contained an evenly spaced 5×5 (synthetic) or 18×18 (*ex vivo*) grid (see example in Figure 1(b)). For the *ex vivo* dataset, we applied in some experiments several augmentations for each epoch using the *albumentations* package (Buslaev et al., 2020): Images were randomly varied in brightness, contrast, gamma and blur. Additionally, images were flipped and rotated. In all experiments, we trained the architecture for 100 epochs using the Adam optimizer together with a constant learning rate of $10^{-3}$. The training and inference was performed on a machine equipped with an NVIDIA RTX 2080 Ti (11 GB) graphics card.

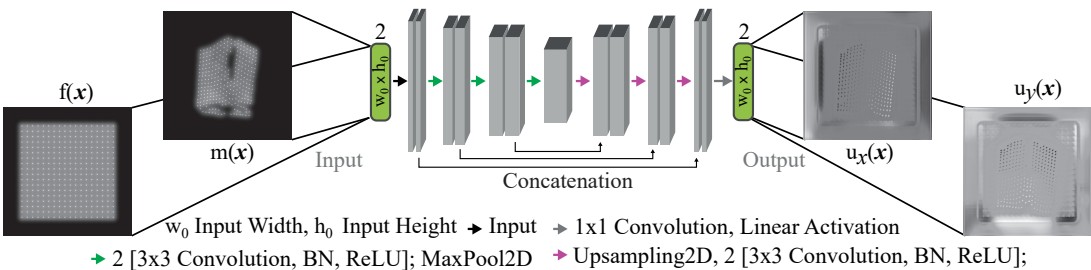

Figure 2: Example neural network architecture for feature-based image registration predicting displacement maps $u_x$ and $u_y$ using a fixed $f(\mathbf{x})$ image and a moving $m(\mathbf{x})$ image (configuration I2 from Table 1)

## 2.5. Displacement maps and feature-based loss function

For each pixel $k$, warped coordinates $x_w^k$ and $y_w^k$ are computed as a function of the displacement maps $u_x$ and $u_y$ and the coordinates of the input image $x_m^k$ and $y_m^k$:

$$x_w^k = x_m^k + u_x(\mathbf{x}^k), \tag{1}$$

$$y_w^k = y_m^k + u_y(\mathbf{x}^k). \tag{2}$$

As no ground-truth deformation maps are available, we use a feature-based metric emphasizing the correspondence of warped and fixed laser rays (keypoints). We therefore minimize the distances between warped and fixed keypoints by knowing the exact location in the grid $x_f^k$ and $y_f^k$. We first compute the Euclidean distance $d^k$ for each keypoint $k$ (Equation (3), in $px$), compute the mean squared Euclidean distance (MSED) for each image for $n$ keypoints (Equation (4), $px^2$) and for each batch consisting of $N$ images (Equation (5)).

$$d^k = \sqrt{(x_w^k - x_f^k)^2 + (y_w^k + y_f^k)^2} \tag{3}$$

$$MSED = \frac{1}{n}\sum_{k}^{n}(d^k)^2 \tag{4}$$

$$\epsilon_{MSED} = \frac{1}{N}\sum_{i}^{N}MSED_i \tag{5}$$

Taken together, we minimize the MSED across the batch to train the network to predict highly accurate, feature-based displacement maps.

## 2.6. Grid classification and evaluation metric

After the image registration, the points were assigned a grid position using a heuristic similar to a nearest neighbour search. As our task is bijective, i.e. one keypoint can only be assigned to one grid position, we assign the keypoint to that position in the grid where it is

globally the closest. This iterative search results in a unique assignment map. We provide a structogram in the Appendix C.1. After we observed that in some cases the points were not assigned in order, we used prior knowledge, e.g. that 3 follows 2 and that 2 does not follow 3, to implement a custom bubble sort algorithm to ensure that points are sorted logically in the grid. For evaluation, we investigate how close the registered points are to their respective, ideal grid location using the mean Euclidean distance in *px*. We further determine the assignment accuracy, i.e. the fraction of points that were assigned correctly in the grid.

## 3. Experiments and Results

Using the generated images, we first investigated if intensity-based algorithms, such as ANTs, were able to closely register any frame to the plane laser grid image. We found that the results were very insufficient (see Appendix A.1 for exemplary image). We therefore asked if a deep neural network-based approach is suitable for a feature-based image registration.

### 3.1. An encoder-decoder network is able to learn feature registration using displacement maps

We next investigated which transformations could be learned using a synthetic dataset of 25 keypoints arranged in five rows and five columns (see also 2.1). The network should be especially robust to highly non-linear, i.e. non-rigid, transformations and missing keypoints, as this is common in endoscopic footage. We therefore tested a series of transformations (Table 1, T1-T3).

| Id | Deformation | Input Images | Figure | |
|----|-------------|--------------|--------|------|
| T1 | **Affine** | Moving | 3(c) | |
| T2 | **Affine + Non-linear** | Moving | 3(c) | |
| T3 | **Affine + Non-linear + Dropout** | **Moving** | 3(c) | 3(d) |
| I2 | Affine + Non-linear + Dropout | **Moving + Fixed** | | 3(d) |
| I3 | Affine + Non-linear + Dropout | **Moving + Difference + Gradient** | | 3(d) |

Table 1: Overview of different training settings for training the synthetic dataset.

As shown in Figure 3(a), all tested transformations can be learned by our network architecture. We observe after roughly 50 epochs a steady state where the loss marginally decreases in the training and validation dataset. As the fixed image is constant across experiments and individual images, we found that only feeding the moving image is sufficient for network convergence. Intruiged by previous studies (Fan et al., 2019; Jaderberg et al., 2015), we tested different input strategies to improve the network performance. We found that feeding more information is in general beneficial for network performance (Figure 3(b)). The median of the mean Euclidean distance (MED) in the laser grid for each keypoint was 1.41 px, 0.46 px and 0.68 px (T3, I2 and I3, respectively), where condition I2 yielded the best results and the narrower distribution (Figure 3(c)). We also tested in our studies the loss (MSED vs. MED) and found equal convergence behavior. Regularization ($L_1$ or $L_2$) impaired the network convergence and yielded worse results. In summary, we found that

the assignment accuracy was across multiple settings at around 97-99%. In Appendix D.1, we provide an overview of the registration accuracy across images.

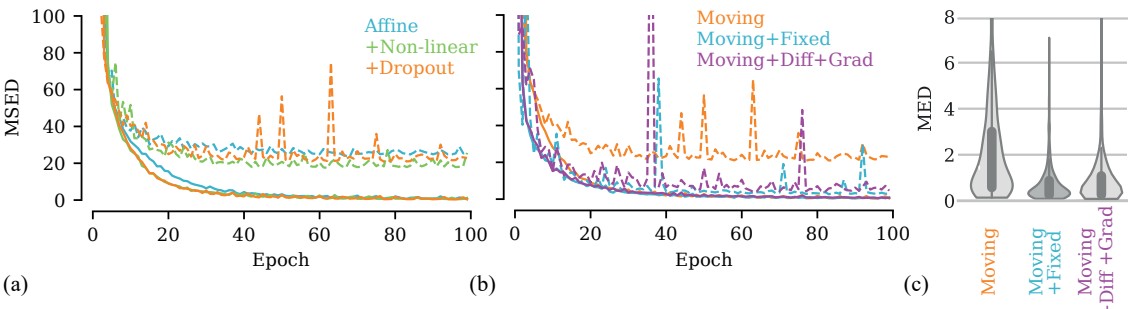

Figure 3: Feature-based registration on a synthetic dataset. (a) MSED loss of training (solid line) and validation (dashed line) set across training epochs for different transformation settings (T1-T3). (b) MSED loss of training (solid line) and validation (dashed line) set across training epochs for different inputs (T3, I2, I3). Same y-axis as in panel (a). (c) Distribution of mean Euclidean distances (MEDs) for different inputs on the validation set (T3, I2, I3).

## 3.2. Feature-based registration performs well on real, highly non-linear deformations

We next investigated if larger image sizes, more keypoints and under real circumstances, the network still performs equally well compared to the synthetic dataset. We used *ex vivo* recordings of calf larynges where an 18×18 laser grid was projected onto and extracted the keypoints manually (see Methods and Figure 1(b)). Similarly, we applied the same architecture as described for the synthetic dataset and trained for 100 epochs (Figure 4(a)). We found that training only on the ground-truth data revealed a high median MED on the validation set and a low assignment accuracy (see 2.6) of 51% on the validation dataset (Figure 4(b,c)). We further evaluated the network's performance on the validation set if purely trainined with adequate synthetic data (same resolution and grid dimensions, see 2.1). Here, we also yielded high median MEDs and a low accuracy of 46% (Figure 4(b,c)). Interestingly, when combining the ground-truth and the synthetic data, we leveraged the performance and gained lower median MEDs of 9.8 px and an accuracy of 72%. We found the largest boost when applying intense augmentations (see 2.4) and increasing the variety 20-fold. In this case, the MED variance was very low (Figure 4(b)) and the MED value at 5.13 px. As the grid spacing is 16 px, this suggests an accurate nearest neighbour sorting. Indeed, the accuracy is around 91% on the validation set (Figure 4(c)). Noteworthy, the highest uncertainty in prediction is where keypoints are missing close to the glottis (see Figure 1(b) and Appendix E.1).

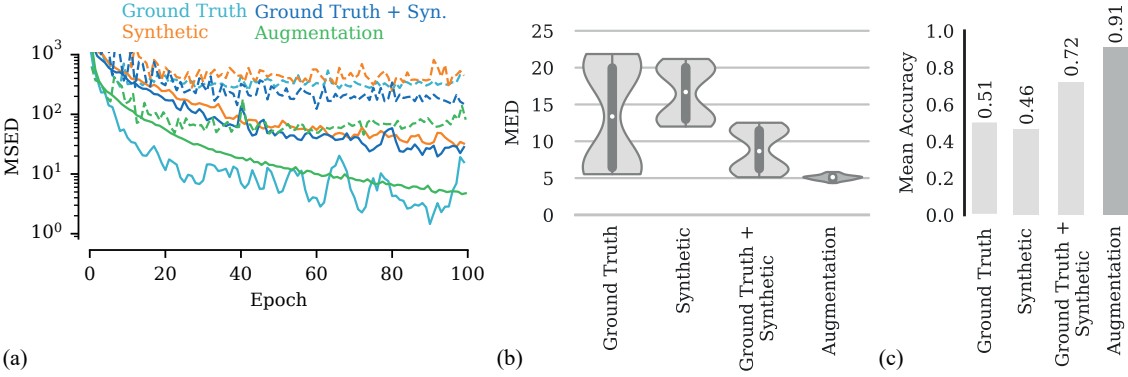

Figure 4: Feature-based registration on an *ex vivo* dataset. (a) MSED loss of training (solid line) and validation (dashed line) set across training epochs for different input data. (b) Distribution of mean Euclidean distances (MEDs) on the validation set for different input data. (c) Mean accuracy on validation dataset.

## 4. Discussion and Conclusion

In this study we suggest a U-Net-like architecture that uses a moving and a fixed image of laser points in rectangular grid, respectively, to compute a deformation field to register keypoints based on their identity. We show that the network is able to learn affine and highly non-linear transformations, and is capable of coping with a large fraction of missing keypoints. However, we have not systematically addressed yet how many missing keypoints can be tolerated by our approach. We found that a larger gap of keypoints result in higher assignment variation, whereas we still found high assignment accuracies of over 91%, which is only slightly lower compared to our toy dataset (97-99%).

We also found that training solely on synthetic data is almost as good as training on only ground-truth data, and a blend of synthetic and ground-truth data enhances the registration and thus, the assignment accuracy (Figure 4). Still, we believe that the non-linear transformations in the *ex vivo* data are not fully represented in our synthetic dataset. Further investigations about the non-linear transformations may help in developing better strategies to generate more realisitc synthetic data.

The data used in this study was manually annotated to evaluate the core idea of using feature-based registration in structured light endoscopy. The extraction accuracy may also have an impact on the registration, as one could potentially miss and/or identify additional keypoints at wrong locations impacting the registration and the assignment. In the future, we will address the complete workflow to uncover sources of error propagation.

In summary, our results suggest that our presented feature-based registration method is highly valuable in structured light endoscopy, such as 3D laryngeal endoscopy, and together with keypoint extraction a potentially fully automatic data analysis technique.

## Acknowledgments

Andreas M Kist was supported by a fellowship of the Joachim Herz foundation. Part of this work was supported by the German research foundation (DFG) under the grant no DFG DO1247/9-1

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

## Appendix A. Intensity-based image registration

We used ANTs with various settings to register the extracted keypoints to the fixed image (regular laser grid). However, no settings resulted in satisfying results (see Figure A.1).

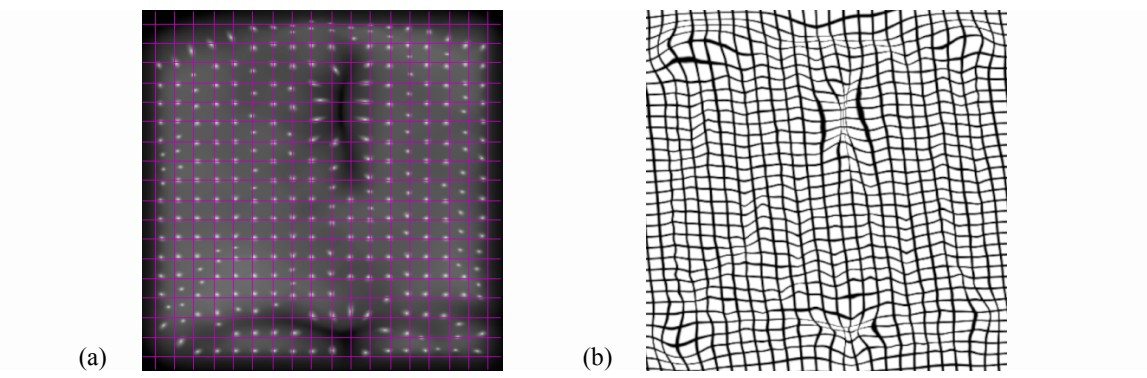

(a)    (b)

Figure A.1: Failed registration using ANTs. (a) morphed image, (b) deformation field.

## Appendix B. Synthetic data generation - example images

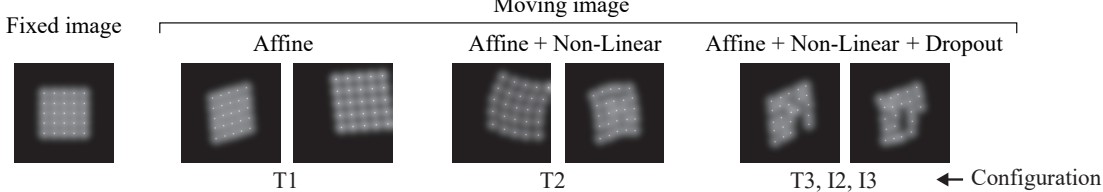

Figure B.1: Synthetic dataset. Fixed image and examples of images with affine transformation (used in configuration T1), with affine and non-linear transformation (T2), and with affine and non-linear transformations with random dropout (T3, I2, I3).

## Appendix C. Bijective nearest neighbour search

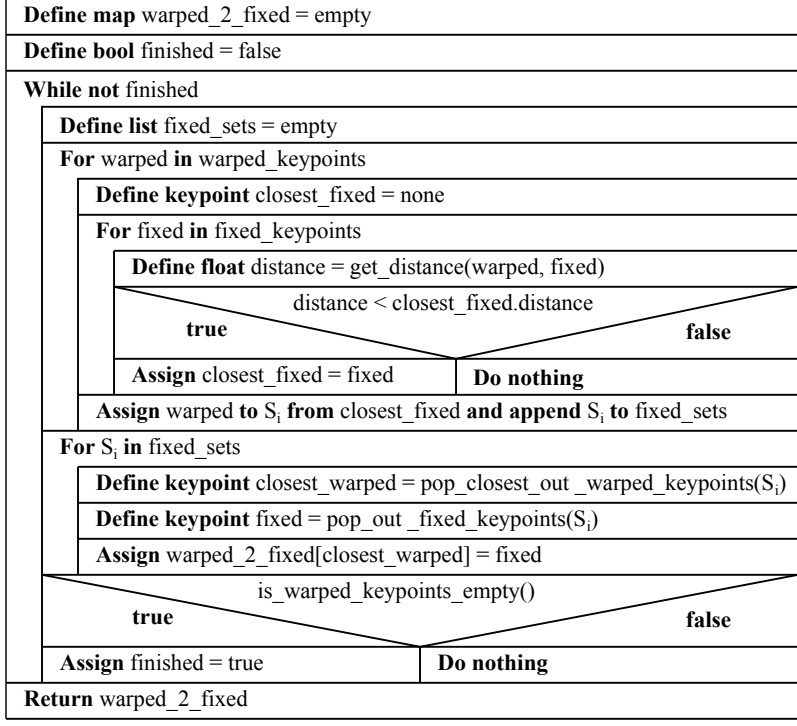

Figure C.1: Structogram of bijective nearest neighbour search

## Appendix D.  Registration accuracy in the synthetic test dataset

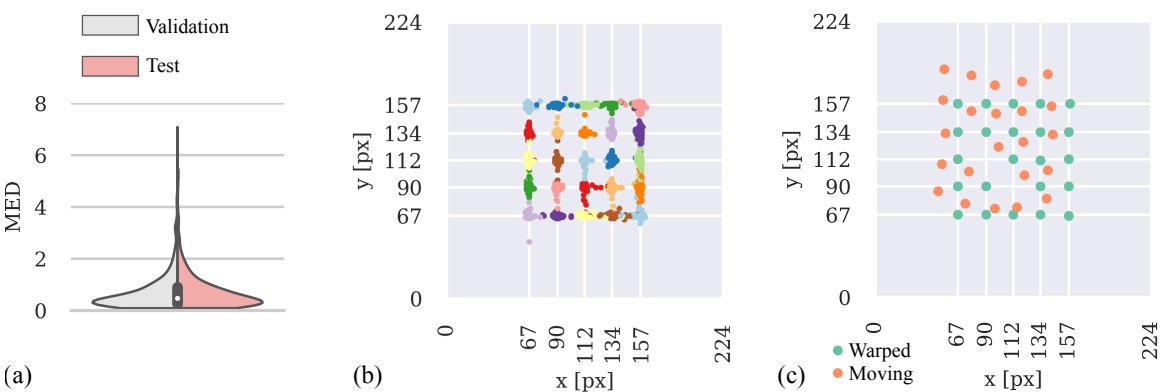

Figure D.1: Registration accuracy in the synthetic test dataset. (a) MED distribution of validation (gray) and test (red) dataset. (b) Registration accuracy across images, color-coded for each keypoint. (c) Example registration of moving image (red dots) to fixed image (white grid). Warped keypoints in green.

## Appendix E. Registration accuracy in the *ex vivo* test dataset

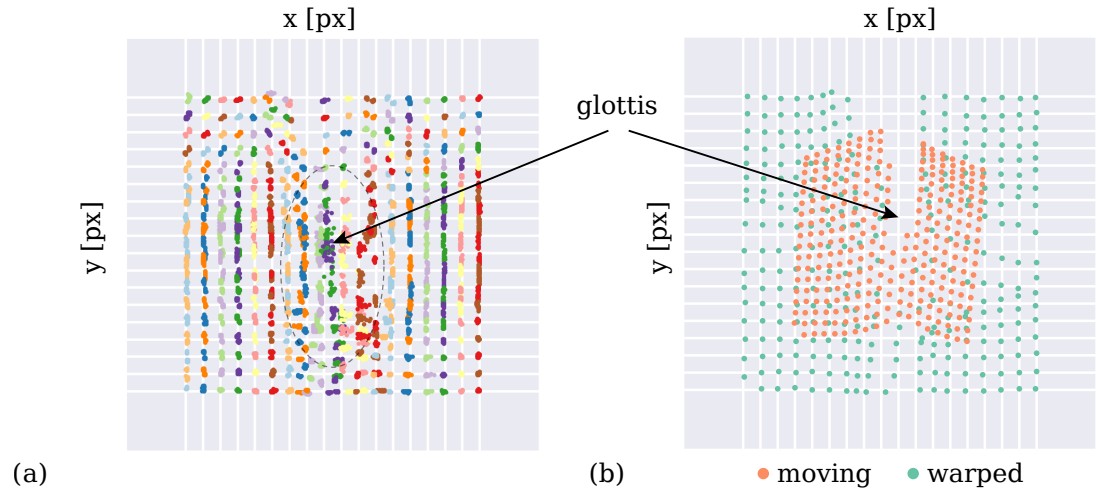

(a)  (b)  ● moving  ● warped

Figure E.1: Registration accuracy in the *ex vivo* test dataset. (a) Registration accuracy across images, color-coded for each keypoint. Note the uncertainty in the center. There, the moving glottis is located and many keypoints are (at least partly) missing. (b) Example registration of moving image (reds dots) to fixed image (white grid). Warped keypoints in green.

