# OpenReview forum: "Feature-based image registration in structured light endoscopy"
_MIDL.io/2021/Conference — MIDL 2021_

### Official Review · ~Alessa_Hering1 · 2021-03-06

**Confidence:** 3
**Preliminary Rating:** 2
**Recommendation:** Poster
**Final Rating:** 3

**Summary:**

The manuscript presents a deep-learning-based image registration approach for aligning two keypoint images. The application for this approach is 3D reconstruction for structured light endoscopy. The extracted laser ray points are registered to a regular grid. The approach is evaluated on synthetic and ex vivo data showing promising results.

**Strengths:**

- The paper is well written and the motivation is clear.
- The paper gives a detailed background.
- the proposed method is tested on synthetic and ex vivo data showing promising results
- Visualization via violin plots are helpful


**Weaknesses:**



After reading the paper, several points are still unclear to me:

- The grid classification section is not clear to me. “the points were assigned a grid position using a heuristic similar to a nearest neighbor search” and “after we observed that in some cases the point were not assigned in order, we used prior knowledge to implement a custom bubble sort algorithm […]”
1. How is it done?
	2. What prior knowledge was used?
	3. All necessary information to understand the method should be in the main paper and not in the appendix.
	4. How do you handle missing keypoints?

- In the abstract, the authors say that they “first separate the laser rays from the background using semantic segmentation”. However, later they say “all visible keypoints were annotated manually”. How are the keypoints extracted? If a segmentation approach is used which one?
- Why is not a point-based registration approach used? Using the keypoints as a point coordinate and align those instead of making an intensity image out of it.
- “We further evaluated the network’s performance on the validation and the test set if purely trained with adequate synthetic data” (p.7) When are the synthetic data adequate? Is the same image resolution used for the synthetic data as for the ex-vivo data in this experiment? If the synthetic data from the first experiments are used: Why weren’t synthetic images with the same resolution and number of keypoints as in the ex vivo dataset used?
- “[…] is capable to cope with a large fraction of missing keypoints” (p. 7) How many keypoints were missing? Does the number of missing keypoints affect the registration accuracy of the other keypoints?
- Why is the registration result of the ANT registration (A1) a failed registration? What do the results of the presented method look like in this picture?


**Deanonymize Review:**

yes

**Detailed Comments:**

- Split of dataset: 10+2+2 = 12. I assume it should be training 8, validation 2, test 2 (p.3)
- The transformer networks of Jadeberg are not applied on a biomedical image registration task (p. 3)
- The result section should be renamed to Experiments and Results – because the experiments were described as well.


**Final Rating Justification:**

The changes made during the rebuttal yield an improved version of the paper. The authors clarified open questions and therefore, the paper is easier to follow.

**Justification Of The Preliminary Rating:**

The paper seems to present an interesting approach and shows promising results but there are several points which are unclear to me after reading the paper (see above). These questions should be answered before accepting the paper.

**Paper Type:**

methodological development

**Questions To Address In The Rebuttal:**

- The paper seems to present an interesting approach and shows promising results but there are a lot of open questions after reading the paper (see above). These questions should be answered.

- Please add a discussion section to the paper and discuss the results, weaknesses, failure cases etc.


**Special Issue:**

no

---

### Official Review · AnonReviewer1 · 2021-03-08

**Confidence:** 4
**Preliminary Rating:** 2
**Final Rating:** 3

**Summary:**

This work seeks to register a laser grid map (projected on to images) to a template grid map. The authors experiment using both a traditional image registration method (the ANTs tool) and a deep learning-based (based on U-net) registration method with a custom loss function to perform this task. Results show that the deep learning method can recover the mapping of grid points to the template.

**Strengths:**

•	The novel loss function to minimize the grid keypoint displacements (from manually identified landmarks) is interesting and appears to work well to train the deep learning registration network.
•	Experiments test different sets of inputs and used for the deep learning algorithm.
•	Results provided for both sythentic data and ex vivo data.


**Weaknesses:**


•	Apparent reliance on manual keypoint extraction (segmentation) for U-net moving image input appears to limit practical application.
•	No details were provided regarding the parameter settings for the traditional registration method, ANTs.


**Deanonymize Review:**

no

**Detailed Comments:**

This is an interesting deep learning approach to grid template matching based on keypoint matching (in this case the keypoints are laser points projected onto an image), and optimization of the mapping coming from a novel loss function that optimizes ground-truth keypoint displacement relative to the known template grid.


Sec. 2.3: My main concern with the proposed method is in the generation of the images used as moving image input to the U-net registration network. It is unclear from the text, but it appears that the laser keypoints require manual annotation to create the input images. I understand that the manual keypoint identification is necessary for the loss function and evaluation, but it appears that they are also used to create the input images to the network. From my reading of the paper, it appears that the extracted keypoints are overlaid onto a smoothed version of the moving image. However, this method does not seem practical in the real world where manual annotation would be time consuming, instead the raw image with noisy laser points should be the input (or include an automated segmentation procedure?). Clarification about this would be helpful.


Sec. 3: What are the registration parameters for ANTs? What intensity metric was used? Additional details about this registration procedure would be helpful for readers to understand why ANTs failed so badly. Without this information, the experiments using ANTs cannot be interpreted well.


Minor comments:

Fig 1(c): I think the rows should be labeled as “n_r” in the figure instead of “n”.


Fig 1(c): It would be helpful for readers to include a legend showing the depth information, e.g. red is high distance values and green low distance.


Sec. 2.5: To improve clarity, I think this subsection could be renamed to something “Displacement maps and keypoint loss function” to better reflect the content under this heading.


Sec. 3.2:  From the text, it is not very clear what is meant by “accuracy” in this evaluation? I think that this is the results of the grid classification of Sec. 2.6, but it is not completely clear in Sec. 3.2. Maybe some text to clarify that this does refer to Sec. 2.6 would be helpful.


Grammatical:
Abstract: “In future” – “In the future”
Sec. 1, p2: “there is a thrive” -> “there is a drive”
Sec. 4, p7: “is capable to cope” -> “is capable of coping”


**Final Rating Justification:**

Thank you for your clarifications and your revisions have improved the quality of the manuscript. While I do feel that standard, non-deep learning registration metrics may still have utility for this problem, the deep learning solution presented here is interesting and of interest to MIDL.  I have upgraded my rating.

**Justification Of The Preliminary Rating:**

While this is interesting application area, the paper in its current form lacks methodological details to determine if the method operates in a fully automated manner at test time, which has the potential to limit practical application.

**Paper Type:**

both

**Questions To Address In The Rebuttal:**

Clarification about what the input data to the U-net really is would be helpful.

What parameter settings were used for the ANTs method? For example, what loss function was used? And other parameter settings would be helpful.


**Special Issue:**

no

---

### Official Review · AnonReviewer4 · 2021-03-08

**Confidence:** 4
**Preliminary Rating:** 2
**Recommendation:** Poster
**Final Rating:** 3

**Summary:**

In this manuscript, a method is proposed to register laser grids and subsequently identify each laser ray to reconstruct a 3D endoscopic view of the larynx. The registration task is performed by training a convolutional neural network to estimate the spatial displacement, followed by custom laser ray classification. The topic of this work is interesting and the application is original. However, certain aspects of this work could be improved (see “weaknesses”).

**Strengths:**

The topic of this work is interesting: the task of registering a regular grid is a challenging image registration task, although I am not fully convinced that classical “intensity-based” algorithms such as ANTs cannot perform this task at all. Would it be possible to develop a feature-based, iterative registration method based on such a classical toolbox? The combination of synthetic data with a small ex vivo dataset is interesting in this context, as this combination seems to significantly boost the network’s performance, especially when combined with data augmentation.

**Weaknesses:**

There are many aspect of the method that are unclear and confusing. Specifically, some details are lacking about the dataset description, data augmentation and (most importantly) experimental setup. There are also some inconsistencies about the reported numbers.

**Deanonymize Review:**

no

**Detailed Comments:**

Throughout the manuscript there are several small mistakes or vague aspects:
- The description of the synthetic dataset in section 2.3 is very limited and needs more detail.
- What do the authors mean with “intense augmentations” (Results section, page 7)?
- What is the role of the validation and the test sets in this study? The authors do not specify how they optimized the hyperparameters of the CNN. Sometimes the results are only calculated for the validation set (e.g. Figure 4b) and sometimes only for the test set (e.g. Figure 4c).
- Section 2.2: “Out of the twelve videos, ten videos belonged to the training, two to the validation and two to the test set.” So… how many videos were there?
- The authors mention an accuracy of 99% on the synthetic dataset in the conclusion section but do not mention this number anywhere in the Results, nor do they explain how they obtained this number.
- The relation between Figure 2 (the network architecture and the inputs/outputs) and Table 1 is unclear. And how do the settings of Table 1 correspond to the results in Figure 3? E.g., are the MEDs presented in Figure 3c based on affine, non-linear and/or dropout deformations? And why are the results shown for the validation set and not for the test set?
- Overall, there are many small English grammar/linguistic mistakes.

**Final Rating Justification:**

In my view, the authors have improved the manuscript based on mu comments.

**Justification Of The Preliminary Rating:**

The proposed methodology and application are interesting and relevant, however, the lack of details about the experimental setup and inconsistencies in the reported numbers do not inspire confidence in the presented work.

**Paper Type:**

methodological development

**Questions To Address In The Rebuttal:**

The authors should primarily address the lack of details and the inconsistencies in the text.

**Special Issue:**

no

---

### Official Review · AnonReviewer3 · 2021-03-09

**Confidence:** 4
**Preliminary Rating:** 4
**Recommendation:** Oral

**Summary:**

The authors present a new automatic method built on imaging registration to correctly identify and assign each laser ray to its respective position in the laser grid. The authors proposed the use of feature-based custom loss and a deep neural network by means of a U-Net-like architecture, instead of intensity-based image registration. They test the new method with in-silico data and one experimental dataset, achieving an accuracy of 91%.

**Strengths:**

The manuscript is well written and very well presented. The authors propose a new method and test it not only with in-silico data but also with the experimental one.  And the results are clearly addressed. They provide also results on the annex


**Weaknesses:**

Although the methodology and results are well described,  the state-of-the art and discussions, need to be improved. Some details are asked in the "question to address" section below
Additionally, I provide some minor edition mistakes in the "Detailed Comments" section below, too

**Deanonymize Review:**

no

**Detailed Comments:**

1) Please define ANTs before using it in the abstract, and CMTK before using it in the introduction.
2) Please, in Figure 1 use the notation nr for the number of rows according to the caption figure.
3) All through the manuscript there are several edition typos such as (reviewed in Döllinger et al. (2001)), Please, remove the parenthesis at (2001)

**Justification Of The Preliminary Rating:**

The manuscript is addressing a needed problem in imaging registration. It presents a new method and provides fairly results. Some minor and major questions are still needed but I really believe for the quality of the work, that the authors are able to address them in the  MIDL timeline

**Paper Type:**

methodological development

**Questions To Address In The Rebuttal:**


1) For the first paragraph in p.3, please re-discuss it by adding state-of-the ART on registration methods. For instance, the following reference or similar works:
Please, see the reference RODRÍGUEZ-RUANO, Alexia, et al. PET/CT alignment for small animal scanners based on capillary detection. En 2008 IEEE Nuclear Science Symposium Conference Record. IEEE, 2008. p. 3832-3835.
2) “The median of the mean Euclidean distance (MED) in the laser grid for each keypoint was 1.41, 0.46, and 0.68” Are they mm?
3) Discussion needs to be improved: The method provided better performance in experimental data than in-silico one, please argue why do you think is like this?. What is the imaging resolution? and comparing the resolution with the MED, how are the percentages differences between methods varying


**Special Issue:**

no

---

### Meta-Review · Area_Chairs · 2021-03-29

**Recommendation:** Accept (Poster)

**Metareview:**

The authors propose a method to register a laser grid to a template grid map, and analyze both classical approaches (via ANTs) and learning-based methods. The task is interesting and somewhat more rarely tackled in itself in the field (to my knowledge), and the authors use synthetic data.

Most reviewers had issues with the clarity of the paper and were perplexed at why ANTs did not perform better (or lacked the necessary details to understand the reason). In addition, the reviewers had a wide range of questions about the various parts of the method, such as how the synthesis and augmentation was done, how the dataset was split, details on results, etc. Overall, the paper could be improved in terms of writing. I strongly encourage the authors to work on these sort of aspects before submission, MIDL enabled for an environment where the authors were given the chance to substantially improve the manuscript and for reviewers to adapt, but this is rare.

Given the discussions, the manuscript improved and all reviewers adjusted to accept. I believe the paper is overall a borderline submission but agree that the updated manuscript would be interesting at MIDL.


**Paper Type:**

methodological development

---

### Decision · Program_Chairs · 2021-03-31

Accept